# Comparing Masticatory Performance of Maxillary Mini Dental Implant Overdentures, Complete Removable Dentures and Dentate Subjects

**DOI:** 10.3390/jcm10215006

**Published:** 2021-10-27

**Authors:** Luc Van Doorne, Ben De Backer, Carine Matthys, Hugo De Bruyn, Stefan Vandeweghe

**Affiliations:** 1Oral and Maxillo-Facial Surgery, Bilkske 68 “Het Tandplein”, 8000 Brugge, Belgium; 2Oral and Maxillo-Facial Surgery, University Hospital, Corneel Heymanslaan 10, 9000 Ghent, Belgium; 3Oral and Maxillo-Facial Surgery, General Hospital AZZENO, Kalvekeetdijk 260, 8300 Knokke, Belgium; 4Oral Health Sciences, Ghent University, Corneel Heymanslaan 10, 9000 Ghent, Belgium; bbacker.DeBacker@ugent.be (B.D.B.); Carine.Matthys@ugent.be (C.M.); Hugo.DeBruyn@ugent.be (H.D.B.); 5Department Periodontology and Oral Implantology, Institute of Health Sciences, Radboud University Medical Centre, 6525 GA Nijmegen, The Netherlands

**Keywords:** mastication, mini dental implants, maxillary overdenture, oral health related quality of life

## Abstract

Background: Mini dental implant (MDI) overdenture rehabilitation for the edentulous maxilla is a valuable, less invasive and affordable treatment alternative for complete removable dentures (CRD). However, comparative quantification of masticatory performance in different oral conditions are scarce. Purpose: This study compares objective masticatory performance of dentate groups, maxillary CRD and MDI overdentures and subjective masticatory performance in maxillary CRD and MDI overdentures. Materials and Methods: Four groups were defined, age 20+ dentate dental students (DS), age 50+ complete dentate subjects (DP), age 50+ maxillary CRD or MDI overdentures/dentate mandible. Objective masticatory performance was evaluated by measuring circular Variance of Hue (VOH) or the mixture degree of two-color chewing gum (Hue-check View Gum^®^ Test). Additionally, subjective masticatory comparison was investigated in the CRD and MDI groups, with a visual analogue scale (VAS) for different food consistencies and the Oral Health Related Quality of Life (OHRQL) OHIP-14 questionnaire. Results: The mean VOH was 0.11 (SD 0.50, range 0.05–0.27) for the dentate dental 20+ students, 0.13 (SD 0.08, range 0.03–0.31) for the 50+ dentate group (*p* = 0.774), 0.41 (SD 0.41, range 0.14–0.76) for the CRD group and 0.39 (SD 0.18, range 0.07–0.76) for the MDI group (*p* = 0.725). Based on the VAS scores, no improvement was found between the CRD and the MDI overdenture group (*p* > 0.050). The mean OHIP-14 total score was 12.10 (SD 15.87, range 0–56) for CRD, while the MDI group (*p* = 0.039) saw a significant improvement to 2.85 (SD 2.85, range 0–15). Conclusion: Comparable results in objective masticatory performance were registered in dentate 20+ and 50+ subjects with a remarkable inferior outcome for the CRD and MDI group. Compared to CRD, MDI overdentures revealed no substantial improvement in subjective and objective masticatory performance. However, for MDI a significant increase in OHRQL was apparent.

## 1. Introduction

Maxillary edentulous patients with dentures and dentate mandibles are commonly encountered in dental practice [1]. Over a prolonged period of time, progressive bone resorption of the anterior edentulous maxilla could occur, defined as “Kelly Syndrome” [2]. To overcome this problem, one could implement implants, which would decrease the compressive force on the mucosa and underlying bone. This would reduce or prevent vertical and horizontal alveolar bone loss [3]. However, in case of advanced maxillary resorption, the absence of adequate bone volume may impede standard diameter (>3.5 mm) implant installation and, therefore, requires additional bone grafting procedures. The rehabilitation of the distal edentulous maxillary area is often a problem due to the pneumatization of the maxillary sinus, while the anterior maxilla, as a result of resorption process becomes far too narrow (knife edge) [4]. Different maxillary sinus augmentation procedures [5] combined with onlay bone grafts in the maxillary frontal region have been used to obtain sufficient bone volume. Beside the autologous bone, as the gold standard, several biomaterials either used solely or combined with the autologous bone and additionally protected by different kinds of resorbable or non-resorbable membranes, preventing early resorption of the graft material [6], have been used but have also heavily complexified the standard implant treatment. Patients’ compromised health status, fear for reconstructive surgery and financial barriers are often encountered as the main reasons for implant treatment withdrawal. For these reasons, the elderly also refuse conventional implant treatment [7]. If the intention would be to select the simplest, least invasive, and least complicated procedure to rehabilitate the edentulous atrophic maxilla, the use of one-piece mini dental implants (MDI) to sustain overdentures could be considered as a valuable alternative. MDI can avoid sinus augmentation or bone reconstruction and are more favorable due to decreased surgical time, morbidity, and better cost-effectiveness. One-piece implants also circumvent the problem of marginal bone resorption induced by screw retained abutments [8]. MDI or “Mini-implants” are defined as implants with diameter of less than 2.5 mm, a subcategory of narrow-diameter Implants reported by Klein et al. [9] and the ITI consensus statement of 2018 [10,11]. However, maxillary MDIs demonstrate an increased risk of failure [12], the reduced costs, minimally invasive surgical procedure and the improved Oral Health Related Quality of Life, especially in the psychological domain [13], could highly compensate this inconvenience.

The primary aim for implant rehabilitation in the edentulous maxillary jaw is the restoration of dental and oral function, including the ability to masticate food [14]. Mastication is defined as the process of chewing food before swallowing and digestion [15]. It differs from masticatory force, which is defined as the maximum biting force on occlusal contacts recorded with strain gauge transducers, piezoresistive transducers, piezoelectric transducers, optical fiber transducers, and pressure-sensitive films [16]. The mastication process consists of two separate processes: the selection of the food placed between the teeth, followed by the breakdown of the particles (breakage) [17]. The ability to masticate can be measured as the masticatory efficiency and masticatory performance [18]. Masticatory efficiency is defined as the effort required to achieve a standard degree of comminution, while masticatory performance is defined as the ability to comminute or mix test food under standardized testing conditions [19]. Until 1950, the evaluation of masticatory performance was only based on the satisfaction of the patients or arbitrary standard accepted procedures [20]. More recently, an increased number of objective tests were developed, such as the comminution technique and the colorimetric technique [21,22]. The first technique is based on the final particle size of the test food after mastication. The smaller the particles size, the better the masticatory performance. The colorimetric technique is based on the mixing degree of a 2-color material (usually color-changeable gum or paraffin wax), determined by visual matching with color scales [23] or digital analysis with a software program [24,25]. Both procedures seemed equal regarding test reliability [21].

One could assume that the provision of MDIs for overdenture support in the edentulous maxilla can restore the masticatory performance, resembling the dentate situation. We must emphasize, however, that a comparative quantification has never been made and cannot be drawn from the literature. The aim of this study was to compare the masticatory performance of 4 distinct groups: dentate dental students over 20 years of age (20+), dentate subjects over 50 years old (50+), 50+ maxillary CRD and MDI overdenture wearers. The following hypothesis (H0) were assessed
Comparable masticatory performance is found in MDI overdenture compared to dentate subjects;Comparable masticatory performance is observed in 50+ dentate subjects with 20+ dental students;Improved masticatory performance with MDI overdentures is found compared to subjects with CRD.

## 2. Materials and Methods

### 2.1. Study Design

Four groups were defined:Group 1:20+ dentate dental students (DS);Group 2:50+ dentate people (DP);Group 3:50+ maxillary CRD/dentate mandible (CRD);Group 4:50+ maxillary MDI/dentate mandible (MDI).

Participants were considered dentate if they had a least five teeth per quadrant present in the maxilla as well as in the mandible. Participants wearing a removable complete denture (CRD) or an MDI overdenture (MDI) could be full or partial dentate in the mandible with natural teeth, an overdenture on implants or a fixed prosthetic restoration. Participants wore their maxillary CRD or MDI overdenture for at minimum 4 to 5 years, allowing sufficient time for habituation.

The treatment protocol and clinical outcome of the MDI patients was described in previous papers [12,13,26,27]. During the yearly follow-up of the MDI clinical prospective study, 20 volunteers were randomly allocated to take part of the masticatory study. The exclusion criteria for all of the groups were temporomandibular joint disfunctions, uncontrolled systemic diseases, severe osteoporosis, bisphosphonate therapy, mental or physical disorders and patients undergoing radiotherapy. The clinical studies were approved by the Ethical Committee of the Ghent University Hospital and registered with the Belgium registration numbers in group DS and DP: BE6702020001105, group CRD: BE6702020000797 and group MDI: BE670201422937. All participants signed an informed consent.

### 2.2. MDI Overdenture Treatment

All of the MDI implants were installed in the maxilla by the same surgeon (L.V.D.), according to a free-handed flapless procedure guided by a preoperative CBCT and the adapted denture used as a surgical guide. The 1-piece MDIs used in this study (ILZ, Southern Implants, Irene, South Africa) were made of high-strength pure grade 4 titanium and had a diameter of 2.4 mm with a length of 10 or 11.5 mm. The MDIs had a machined surface (Sa: 0.4 mm) of 4.8 mm in length at the coronal part and a roughened surface (Sa: 1.5 mm) further apically. Early loading by adaptation of the denture with a soft tissue reliner Coesoft gel (GC America, Chicago, IL, USA) was carried out. After six months a final horse-shoe denture with palatal metal reinforcement was installed. The example of the clinical intra oral situation is demonstrated in Figure 1.

Patients were invited for follow-up on a strict time schedule. Postoperative visits followed 1 week, after 1, 3, 6, 12 and 24 months and yearly until the final 5-year control.

### 2.3. Objective Masticatory Performance Test

Masticatory performance was evaluated by measuring the mixture degree of two-color-changeable gum of the Hue-check View Gum^®^ Test (Orophys, Bern, Switzerland) described and validated by Schimmel M. et al. [25,28] and Halazonetish D. et al. [29]. The objective masticatory performance was investigated in the 4 defined groups. Each subject was instructed by the same operator (B.D.B.) to chew gum on the preferred chewing side for 20 cycles. The chewed gum was put into a transparent plastic bag labelled with date and name of participant and flattened to a 1-mm-thick specimen by pressing it with a roller on a custom-made polyvinyl chloride plate with a milled depression of 1 × 50 × 50 mm. Both sides were scanned using a flatbed scanner (Epson Perfection V600 Photoscanner Model J252A; Seiko Epson Corp., Nagano, Japan) at a resolution of 500 dpi. Segmentation of the flattened gum area was performed with ViewGum software (dHAL Software, Kifissia, Greece, www.dhal.com (accessed on 25 October 2021)) transforming the pixels from red, green, blue (RGB values) to the hue, saturation, intensity (HIS) color space, which is perceptually more relevant to human vision. Mixing the two colors of chewing gum especially modifies the hue. The circular variance of hue (VOH) was considered to be the measure of mixing [30], with a higher VOH value connotating poorer mixing ability of the two-colored layers. If the colors of the chewing gum are not mixed, two well-separated groups with larger VOH can be observed. On the other hand, when mixing is adequate, the two groups approximate with a smaller VOH. Digital View Gum analysis is demonstrated in Figure 2.

### 2.4. Subjective Masticatory Evaluation

Subjective masticatory evaluation was only investigated in the CRD and MDI group. Participants were asked to score their subjective impression and capability to chew a piece of soft white bread, a piece of hard cheese, a piece of dry sausage, a piece of apple and a piece of carrot on a 100 mm Visual Analogue Scale (VAS) [31] (Appendix A). Additionally, they were asked to fill out the validated Dutch version of the OHIP-14 questionnaire (Appendix B), to evaluate the impact of the treatment on Oral Health Related Quality of Life (OHRQoL) [32,33,34]. The latter is a practical instrument in clinical settings [35] and has proven reliability, validity and precision which allows for a comparison between studies [36]. The questions are based on Locker’s theoretical model of oral health [37] with seven formulated dimensions, namely: functional limitation, physical pain, psychological discomfort, physical disability, psychological disability, social disability and handicap. The validated Dutch version (OHIP-NL) [38] of the oral health impact profile (OHIP-14) [36] questionnaire has 5 categories of response for each ite: never (=0), hardly ever (=1), occasionally (=2), fairly often (=3) and very often (=4). The sum of the responses for each item are defined as the Total OHIP-14. The maximum positive score is 0, indicating a high appreciation of quality of life whereas the maximum negative score 56 indicates a low quality of life [39].

### 2.5. Statistic Evaluation

Statistical analysis was performed using SPSS version 25 (IBM SPSS, statistics for Windows, version 25.0, Business Analytics, Amonk, NY, USA). The Mann–Whitney U-test was used to compare the outcome variables in between the different study groups, with the level of significance set at *p* < 0.050.

## 3. Results

The demographic distribution of the four groups is depicted in Table 1.

In the maxillary CRD and MDI group the antagonistic mandibular jaw was characterized by natural teeth in 24 participants, the combination of natural teeth and a removable partial denture in 8 participants and another 8 had a 2-implant overdenture. The mean age of the maxillary dentures in the CRD group was 5.8 years (SD 2.1, range 1–13), while the mean age of the MDI overdentures was 3.2 years (SD 0.8, range 1–4).

The objective masticatory evaluation was performed in all of the subjects of the 4 defined groups. The mean VOH of every group is depicted with boxplots in Figure 3.

Both dentate populations demonstrated a significant better chewing ability compared to the CRD and MDI overdenture group (*p* < 0.001). The chewing ability was not affected by the type of dentition in the lower jaw (*p* = 0.642).

Analyzing subjective masticatory performance with VAS and the OHIP-14 questionnaire was solely investigated in the maxillary removable denture groups (CRD and MDI). 

The VAS outcome for different food consistencies is visualized in Figure 4.

No significant improvement was found between the CRD and the MDI group comparing mastication of progressive tougher food consistency (*p* > 0.050).

On the contrary, improvement in the mean Total OHIP-14 score, expressed by a lower Total OHIP-14 value revealed a significant reduction from 12.10 (SD 15.87, range 0–56) for the CRD group to 2.85 (SD 2.85, range 0–15) for the MDI group (*p* = 0.039). The mean individual domain scores, summed to yield the total OHIP-14 score, are depicted in Figure 5. Significant differences in terms of psychological discomfort (*p* = 0.028), physical disability (*p* = 0.006) and social disability (*p* = 0.020) were apparent.

## 4. Discussion

To our knowledge this is the first objective and subjective evaluation of masticatory performance in different dentate age groups, maxillary edentulous subjects with CRD and maxillary MDI overdentures. Motoric functions of masticatory organs such as the tongue, lips, cheeks, and mandible tend to change with age, thereby influencing the masticatory performance [40]. Hence, we found no statistically significant changes in masticatory performance related to age, comparing complete dentate subjects over 20 years old and over 50 years old. On the other hand, being edentulous in the maxilla with CRD or MDI overdentures reveals a significant perturbation of masticatory function. On the contrary to what one would expect, a better retention of the maxillary denture by means of MDI did not significantly improve the objective mastication. Similar observations were reported when denture adhesives were used [41]. An early study [42] mentioned that this is caused by the reduced sensitivity of the food manipulating organs which gives rise to inefficient manipulation of food during chewing. Mastication is a complex process that requires coordination from different structures. Improving denture retention alone may not be sufficient to improve masticatory performance. Research has demonstrated that denture stability is far more important than denture retention for the mastication process [43]. The influence of the ridge form on denture stability, mastication performance and patient satisfaction is also debatable [44]. Therefore, it is assumed that the masticatory performance depends to a large extent on the patient’s adaptability and salivatory flow [45,46]. Another important feature is the absence of a periodontal ligament in implants which differs substantially from natural teeth. The periodontal ligament provides the central nerve system with feedback for sensory perception and motor control, called proprioception. Conversely, the lack of such proprioception causes lower tactile sensitivity and less coordinated masticatory muscle activity [47]. Moreover, masticatory performance improves with the continuous use of newly inserted removable dentures, which might explain why the test failed to demonstrate a significant difference with the overdenture [48].

An objective evaluation of the masticatory performance is not easy to perform. A systematic review by Tarkowska et al. [49] indicated that the use of color-changeable chewing gum is a valid and reliable method for the evaluation of masticatory function. The View Gum^®^ software program introduced by M. Schimmel is designated as the gold standard method and recommended as a suitable method for subjects with impaired mastication [19,50].

An evaluation of the subjective masticatory outcome comparing CRD with MDI reveals absence of significant improvement of the VAS scores. However, a remarkable improvement in OHIP-14 or OHRQL is apparent in favor of the MDI treatment. J. Feine et al. [31] stated that the evaluation of masticatory performance should be preferentially based on patients’ perceptions. The discrepancies between perceived ability to function and laboratory performance in our study partially confirms these conclusions. An objective outcome is not always the best judge of treatment success for the patient.

Limitations of the present clinical investigation included variables such as orofacial pain, occlusal forces, or the role of other elements such as the tongue or cheeks were not analyzed [51].

In the MDI group longitudinal OHRQoL was already measured starting from baseline with a larger sample of 31 subjects and published in an earlier paper [13]. At baseline, the total OHIP-14 was 21.3 (SD 13.1) with the original denture and the final outcome 6.5 (SD 8.9) after 3 years of MDI overdenture in function. The baseline OHIP-14 of this group could be similarly interpreted to the group with a complete denture, however, we must admit that these dentures were already ameliorated before surgical intervention to use as a surgical guide. Therefore, in this study we incorporated a new control group of complete removable dentures. Comparing the high OHIP-14 baseline value of 21.3 (SD 13.1) in the MDI group with our new CRD group 12.10 (SD 15.87) clearly demonstrates the inclusion of subjects complaining of instability and discomfort of their upper conventional denture for the MDI study treatment protocol.

For the objective masticatory test in the CRD group the quality of the existing denture was not objectively assessed. Hence Carlsson & Omar [52] have reported that this has little effect on patient satisfaction.

Finally, general pathologies such as uncontrolled diabetes mellitus, autoimmune diseases, infections, lupus erythematosus, among others that may influence masticatory function and were not investigated in this study and therefore should be included in future research.

## 5. Conclusions

Using objective masticatory performance test following conclusions were reached:No comparable masticatory performance in maxillary MDI overdentures compared to dentate subjects;Comparable masticatory performance in dentate age 20+ years and age 50+ years;No improved masticatory performance in maxillary MDI overdentures compared to CRD in maxillary edentulous subjects.

However, an objective outcome is not always the best judge of treatment success for the patient. Improved patient perceptions of masticatory performance with MDI overdentures are more important and are partially sustained in our research by the outcome of the subjective masticatory evaluation. No improvement in the subjective masticatory performance with VAS for different food consistencies was observed. Hence, a significant increase in OHRQL with the OHIP-14 questionnaire is apparent.

## Figures and Tables

**Figure 1 jcm-10-05006-f001:**
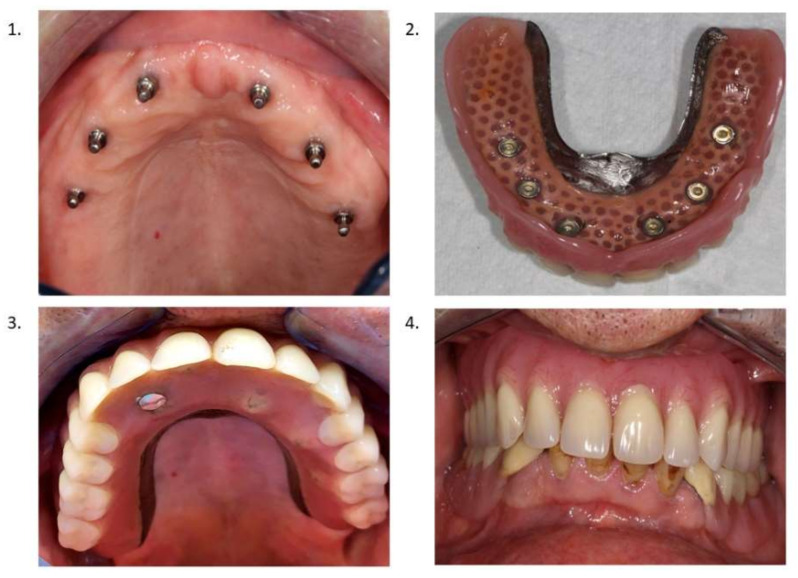
(**1**) MDI in atrophic maxillary jaw; (**2**) MDI overdenture mucosal side; (**3**) MDI overdenture in situ; (**4**) MDI overdenture in occlusion.

**Figure 2 jcm-10-05006-f002:**
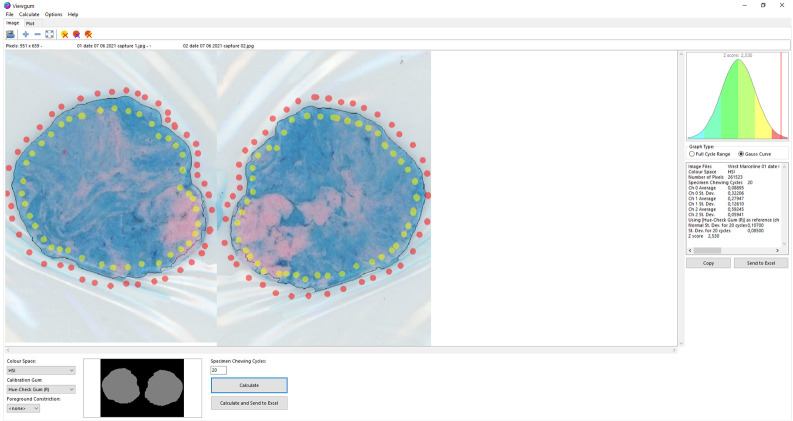
Analysis of flattened two-color chewing gum with ViewGum software (dHAL Software, Kifissia, Greece, http://www.dhal.com/viewgum.htm (accessed on 25 October 2021)) after performing the Hue-check View Gum^®^ Test (Orophys, Bern, Switzerland).

**Figure 3 jcm-10-05006-f003:**
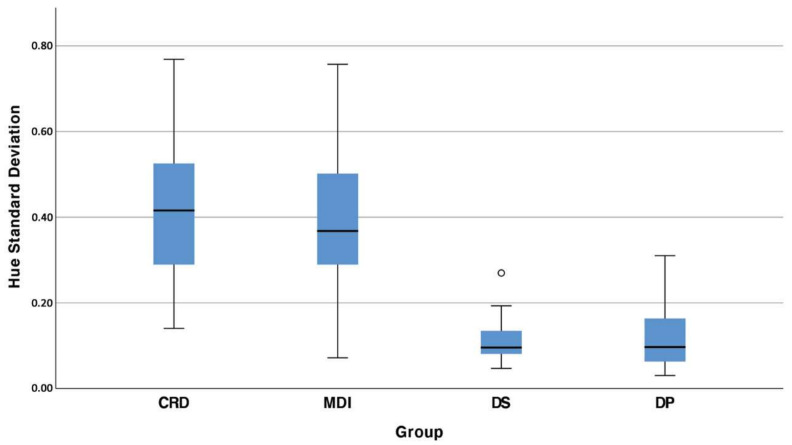
Boxplot of the mean VOH in 4 different groups: (CRD) complete removable denture maxilla/dentate mandible; (MDI) mini dental implant overdenture maxilla/dentate mandible; (DS) dental students; (DP) dentate subjects.

**Figure 4 jcm-10-05006-f004:**
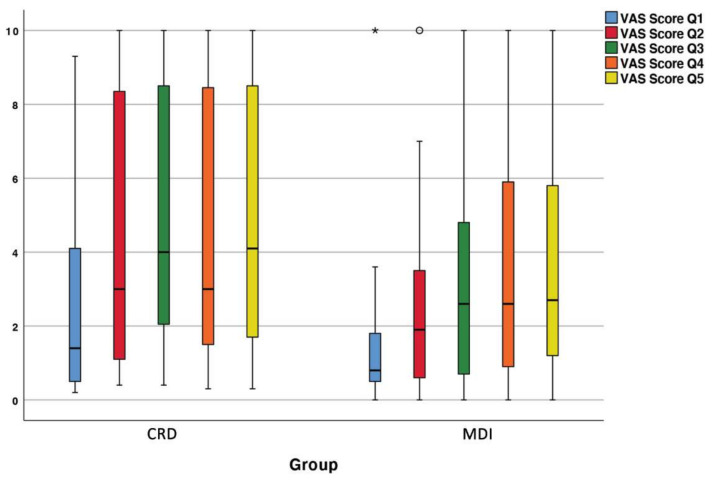
Mean VAS score outcome for masticating food with different con-sistencies (Q1 blue bar: soft white bread; Q2 red bar: hard cheese; Q3 green bar: hard sausage; Q4 orange bar: apple; Q5 yellow bar: carrot) comparing CRD with MDI group. There were no significant differences in the VAS score between both groups (*p* > 0.05). *: Outlier.

**Figure 5 jcm-10-05006-f005:**
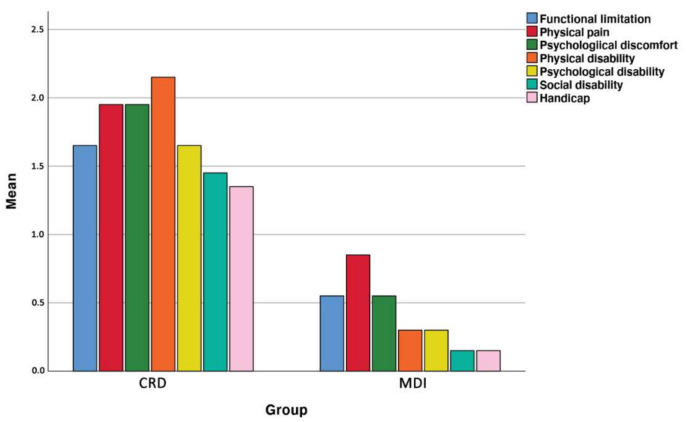
OHIP-14 outcome in different domains in CRD and MDI group. Significant differences were noted for psychological discomfort (dark green), physical disability (orange) and social disability (light green).

**Table 1 jcm-10-05006-t001:** (DS) dental students; (DP) dentate subjects; (CRD) complete removable denture maxilla/dentate mandible; (MDI) mini dental implant overdenture maxilla/dentate mandible.

Group (N)	Mean Age (SD)	Gender (Male/Female)
DS (22)	24.18 (2.37)	10/12
DP (19)	60.53 (8.29)	11/8
CRD (20)	68.4 (6.86)	12/8
MDI (20)	65.75 (8.21)	9/11

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
