# Peer review of "Comparing Masticatory Performance of Maxillary Mini Dental Implant Overdentures, Complete Removable Dentures and Dentate Subjects"

_jcm, 2021, doi:10.3390/jcm10215006_

Round 1
Reviewer 1 Report
It is an interesting paper, however, the way of presenting at results is a bit difficult to understand for readers.
・Please add a figure or table regarding the results of VOH, because the analyses of VOH seem to be a primary analysis of the present study.
・Regarding Figure3 and 4, the authors should add the results of DS and DP groups to consist with the hypothesis. And the authors should clearly indicate which groups were compared using statistical methods at each figure.
・The fourth hypothesis is ambiguous.
Reviewer 2 Report
This article entitled “Comparing masticatory performance of maxillary mini dental 2 implant overdentures, complete removable dentures and dentate subjects.” is aimed to compare objective masticatory performance of dentate groups, maxillary CRD and MDI overdentures and subjective masticatory performance in maxillary CRD and MDI overdenture.
Authors have well revised several issues; however, I ask authors to add some key concepts. Authors must discuss more on the biomechanical factors concerning peri-implant bone resorption ( see reference as Sinjari B, D'Addazio G, Traini T, et al. A 10-year retrospective comparative human study on screw-retained versus cemented dental implant abutments. J Biol Regul Homeost Agents. 2019;33(3):787-797.) and it would also be interesting to include a part on the biomaterials used in bone regeneration that can act as scaffolds for the insertion of the implants themselves (evaluate the pros and cons of the biological mediators involved see doi: 10.1111 / clr.12423) to highlight the difficulties, not only economic, in the insertion of implants in patients with advanced maxillary resorption.
Minor issues:
Conclusions cannot be reduced to a sentence: you must improve them highlighting the limits and the future insights pointed out from this article.
Several moderate typos are present in the text, please, amend.
Round 2
Reviewer 1 Report
There are no comments.